# The Effects of Moderate- to High-Intensity Physical Exercise on Emotion Regulation and Subsequent Cognitive Control in Highly Psychologically Stressed College Students

**DOI:** 10.3390/healthcare13172100

**Published:** 2025-08-23

**Authors:** Baole Tao, Tianci Lu, Hanwen Chen, Jun Yan

**Affiliations:** College of Physical Education, Yangzhou University, Yangzhou 225127, China; dx120190064@yzu.edu.cn (B.T.); dx120240094@stu.edu.cn (T.L.);

**Keywords:** physical exercise, cognitive reappraisal, expressive suppression, cognitive control, psychological stress

## Abstract

**Objectives:** Chronic psychological stress among college students increases sensitivity to negative emotional stimuli. Emotion regulation, a critical coping mechanism, draws upon cognitive resources and may impair subsequent cognitive control. Physical exercise has been proposed as an effective intervention to enhance both emotional and cognitive functioning. This study investigated whether a 12-week structured exercise intervention could modulate emotion regulation outcomes and improve cognitive control in college students experiencing high psychological stress. **Methods:** Forty-seven college students, identified as highly stressed via the Chinese College Students Psychological Stress Scale, were randomly assigned to either an exercise group (*n* = 25) or a control group (*n* = 22). The exercise group participated in supervised rope-jumping sessions three times per week for 40 min, following ACSM guidelines, over 12 weeks. Before and after the intervention, participants completed tasks measuring two emotion regulation strategies—expressive suppression and cognitive reappraisal—and tasks assessing cognitive control. **Results:** A significant group × time × strategy interaction emerged for subjective emotional valence: following the intervention, the exercise group reported attenuated negative valence under expressive suppression. For subjective arousal, post-intervention ratings decreased under suppression but increased under reappraisal in the exercise group, suggesting strategy-specific modulation by physical activity. Regarding cognitive control, electrophysiological measures revealed that the P3 component showed a significant interaction: the exercise group exhibited enhanced P3 amplitudes in congruent versus incongruent conditions after the intervention. Moreover, P3 interference scores were significantly reduced post-intervention in the exercise group compared to both its pre-intervention baseline and the control group. **Conclusions:** A 12-week aerobic exercise intervention enhanced emotion regulation outcomes and improved cognitive control under high psychological stress. These findings underscore the utility of physical exercise as a non-pharmacological approach to bolster cognitive–affective resilience in young adults.

## 1. Introduction

In recent years, the impact of high psychological stress on college students has garnered increasing attention in the field of mental health research. Psychological stress refers to the subjective experience when individuals perceive that situational demands exceed their coping resources. “High psychological stress,” specifically, indicates a chronic or intense experience of stress characterized by persistent feelings of overwhelm, anxiety, and inability to cope effectively with daily academic and social demands. Recent epidemiological studies have demonstrated that college students report experiencing high psychological stress at some point during their academic careers, making it a significantly prevalent issue in higher education settings [1]. Physiologically, chronic high psychological stress has been associated with altered cortisol secretion patterns, increased sympathetic nervous system activity, and disrupted sleep patterns. Psychologically, it often manifests through heightened emotional reactivity, impaired cognitive control, and difficulties in emotional regulation, thereby substantially compromising daily functioning and academic performance. Existing studies have established a strong link between elevated stress levels and individuals’ heightened sensitivity to, and engagement with, negative emotional stimuli [2]. Among college students, such heightened reactivity often leads to excessive focus on negative affect and deep emotional engagement, contributing to a range of maladaptive responses—including anxiety, anger, and fear—that reinforce a negative feedback loop of stress and emotional dysregulation. These patterns present a considerable challenge to students’ emotional and cognitive adaptation. Thus, understanding how students under high stress regulate their emotions—and how such strategies affect psychological functioning—has become a key issue in mental health research.

Emotion regulation refers to how individuals modify the quality, duration, or intensity of their emotional experiences in response to environmental or internal demands [3]. This function is essential for maintaining internal homeostasis, particularly under stress. Two of the most widely studied strategies are cognitive reappraisal—which involves reinterpreting emotional stimuli to alter their psychological impact—and expressive suppression, which involves inhibiting the outward expression of internal emotional states [4,5]. While cognitive reappraisal is generally considered more adaptive, especially in Western contexts, expressive suppression remains culturally common and socially reinforced in collectivist societies such as China. Both strategies are known to modulate emotional experience and neural responses [6]; however, they differ in their cognitive costs and neural dynamics. For example, reappraisal has been shown to reduce the late positive potential (LPP), an ERP component linked to sustained emotional processing.

In contrast, suppression tends to yield less consistent modulation of this component [7]. Neuroimaging studies further reveal that both strategies engage overlapping yet distinct regions of the prefrontal cortex, including the ventrolateral prefrontal cortex (VLPFC), anterior cingulate cortex (ACC), and Broca’s area [8,9]. Notably, younger adults tend to favor avoidance-based strategies, such as suppression, which may reflect developmental limitations in emotional experience and regulation capacity [10]. Moreover, chronic reliance on suppression has been linked to cognitive–affective disorders and may lead to impairments in cognitive control.

From a theoretical perspective, ego depletion theory posits that all forms of self-regulation, including emotion regulation, draw upon a finite pool of cognitive resources [11]. Thus, engaging in regulation strategies—whether through reappraisal or suppression—may deplete cognitive resources and impair performance on subsequent tasks requiring executive control. Resource depletion has been shown to disrupt processes such as information encoding, attentional allocation, and working memory updating [11,12]. The dual competition model further argues that emotional responses, or efforts to regulate them, compete with cognitive tasks for shared neural resources, particularly in the prefrontal cortex [13]. As a result, emotion regulation may incur cognitive costs that undermine subsequent task performance [14].

Physical exercise has emerged as a promising strategy for enhancing cognitive control and reducing the impact of stress. Exercise is a resource-enhancing intervention that may offset the cognitive costs associated with emotion regulation [15,16]. The mechanisms underlying this effect are multifaceted, involving psychosocial, behavioral, and neurophysiological pathways [17]. Psychosocially, exercise improves psychological well-being by enhancing mood, reducing anxiety and depression, and fostering feelings of competence and self-efficacy. Additionally, physical activity facilitates social engagement, providing opportunities for social support and interaction [18], which can buffer stress responses and promote emotional resilience. Behavioral mechanisms such as stress distraction also play a critical role, where physical exercise offers a productive diversion from stressful events, helping individuals regulate their emotions more effectively [19]. Throughout the lifespan, physical activity has been positively associated with cognitive performance. Notably, the prefrontal cortex, which underlies executive function, continues to mature well into early adulthood, with structural development in gray and white matter extending into the third decade of life [20]. Physical exercise has been shown to support this developmental trajectory [21]. Neurophysiologically, regular physical activity promotes angiogenesis, synaptogenesis, and neurogenesis—factors associated with enhanced cognitive capacity. It also increases cerebral blood flow and boosts neurotransmitter levels, improving arousal, attention, and alertness [22]. These physiological benefits may enable more efficient cognitive resource allocation during demanding tasks, especially in emotionally taxing contexts.

High-intensity physical exercise effectively promotes important physiological and psychological stress reduction and cognitive enhancement adaptations. Specifically, prior research indicates that high-intensity physical exercise efficiently activates neurophysiological pathways—such as increased neurogenesis, synaptic plasticity, and neurotransmitter modulation—and reliably improves emotional resilience and cognitive performance under stress [23]. Compared with lower-intensity exercises, high-intensity physical exercise is particularly effective in inducing immediate and sustained improvements in mood, reducing perceived stress, and facilitating executive function, thus offering a suitable intervention strategy for the highly stressed university-student population targeted in the current study [24].

Based on this growing body of evidence, it is hypothesized that physical exercise may influence both emotion regulation and subsequent cognitive control among college students under chronic psychological stress. However, there is currently no direct empirical evidence showing whether exercise improves post-regulation cognitive performance by altering the availability of cognitive resources. To address this gap, the present study employs a direct experimental design comparing two emotion regulation strategies—cognitive reappraisal and expressive suppression—before and after a 12-week exercise intervention. This approach aims to clarify the role of physical exercise in modulating the interplay between emotion regulation and cognitive control, with implications for developing effective interventions for stress-related dysfunctions such as anxiety, depression, and behavioral addiction.

This study proposes the following hypothesis: high-intensity physical exercise may influence emotion regulation and subsequent cognitive control in college students with high psychological stress.

## 2. Materials and Methods

### 2.1. Experimental Design

#### 2.1.1. Participant Screening and Grouping

An online survey of 715 undergraduate students was conducted using a demographic questionnaire and the Chinese College Students Psychological Stress Scale (CCSSPSS). The Chinese College Students Psychological Stress Scale (CCSPSS) was used to assess college students’ psychological stress levels over time. The scale comprises 85 items and five subscales: academic, lifestyle, social, developmental, and family. A 7-point scoring method was used, ranging from 1 (no impact or minimal impact) to 7 (significant impact) [25]. Higher scores indicate higher levels of psychological stress. In this study, the Cronbach’s alpha coefficient for the scale was 0.984. In total, 494 valid responses were collected, yielding a response rate of 69%. The sample comprised 208 males (42.1%) and 286 females (57.9%).

Based on the T-score classification criteria of the CCSPSS, students with stress scores above 61 were identified as experiencing high levels of psychological stress. Eighty-nine participants met this criterion and were considered eligible for further selection [25]. To ensure the reliability and safety of the experimental procedures, participants were further screened according to the following inclusion criteria: (1) no psychiatric disorders or hereditary diseases, with overall good mental health; (2) normal or corrected-to-normal vision (visual acuity > 0.8), with no color blindness or color weakness; (3) no history of substance or alcohol dependence; (4) right-handedness; and (5) no prior participation in similar experimental protocols.

Ultimately, 50 students who met the criteria and voluntarily agreed to participate in a physical exercise intervention signed informed consent forms. Before the formal experiment, the participants received standardized rope-skipping training and were compensated upon study completion.

A priori power analysis using G*Power 3.1.9.7 indicated that a minimum total sample size of 26 was required to achieve adequate statistical power (power = 0.85, effect size = 0.25). During the experiment, one participant withdrew due to physical discomfort, and two were excluded due to poor data quality. The final dataset included 47 participants, with 22 in the control group (11 males, 11 females) and 25 in the exercise group (12 males, 13 females). The average age of the participants was 19.12 ± 1.02 years. Participants were randomly assigned to the experimental or control group using block randomization, stratified by gender to ensure balance across groups. A block size of 4 was used, with the allocation sequence generated using SPSS (Chicago, IL, USA, 26th edition). Group assignments were concealed until the point of assignment. Thus, the final sample size met the requirements for statistical validity.

The participants underwent jump rope training prior to the experiment. Participants were compensated upon completion of the study. Informed consent was obtained from all participants. This study was reviewed and approved by the Ethics Review Committee of Yangzhou University (NO: YXYLL-2022-109). The study was conducted in accordance with relevant ethical guidelines and regulations.

#### 2.1.2. Physical Exercise Intervention Programs and Procedure

The exercise intervention in this study was designed according to the guidelines issued by the American College of Sports Medicine (ACSM) [26]. According to these recommendations, engaging in physical activity 3 to 5 days per week is associated with significant psychological benefits. Moreover, as little as 10 to 15 min of aerobic exercise can yield mood-enhancing effects. In comparison, sessions lasting at least 20 min have been shown to produce more pronounced improvements in anxiety reduction and emotional regulation.

Importantly, prior research indicates that low-, moderate-, or high-intensity aerobic exercise can all positively affect emotional well-being. However, moderate- to high-intensity exercise appears to confer greater psychological benefits, particularly in populations experiencing elevated stress or emotional dysregulation [27].

Therefore, the current study’s intervention was structured to meet these criteria, aiming to optimize physiological efficacy and emotional outcomes in college students experiencing high psychological stress.

In the experimental group, participants engaged in a structured rope-jumping group exercise program in addition to their regular school-based physical activities (e.g., morning exercises, physical education classes). The intervention was conducted three times per week, each session maintaining a moderate-to-vigorous intensity corresponding to 64–76% (moderate) and 77–95% (vigorous) of the participants’ maximum heart rate (HRmax), as defined by the ACSM guidelines.

Each session lasted approximately 40 min, comprising 5 min of warm-up, 30 cumulative minutes of aerobic rope-jumping, and 5 min of cool-down and stretching. The intervention period spanned 12 weeks, during which Polar heart-rate monitors were used to ensure that the participants remained within the target heart-rate zone throughout the sessions (Figure 1). Participants were instructed to perform rope-jumping continuously for 30 min, with the only breaks being self-paced rests during the session, if needed. During the experimental period, the heart-rate range of the subjects was 148.878 ± 10.372 beats per minute. As shown in Figure 2, participants consistently achieved the prescribed cumulative duration and intensity of aerobic activity across the 12-week intervention. The control group participated only in regular school-based physical activities, including daily morning calisthenics (approximately 15 min per session, involving moderate-intensity aerobic exercises and stretching) and mandatory physical education classes twice per week (each lasting around 45 min, comprising moderate-intensity activities such as running, basic sports skills practice, and team games). Participants in the control group were explicitly instructed not to engage in any additional exercise or structured physical training beyond these standard school activities during the intervention period. No supplementary or intensive physical training sessions were arranged for this group.

### 2.2. Experimental Task

#### 2.2.1. Materials

One hundred sixty emotional images were selected from the Chinese Affective Picture System (CAPS) [28], including 40 negative images for passive viewing, 40 neutral images for passive viewing, 40 negative images for cognitive reappraisal, and 40 negative images for expressive suppression. To validate the emotional categorization of these images, an independent group of 10 evaluators was recruited. The evaluators were selected based on the screening criteria for the participants. Each evaluator rated the images on a standardized Likert scale assessing valence and arousal.

The arousal levels were statistically equivalent across image categories (neutral = 5.27, negative = 5.48), while the valence ratings differed significantly (neutral = 5.27, negative = 2.66), confirming the successful categorization of emotional stimuli. The 120 negative images were further divided into three experimental conditions—negative viewing, cognitive reappraisal, and expressive suppression—with each condition comprising 40 images. There were no significant differences in either valence or arousal levels across these three sets of negative images, ensuring comparability between conditions.

The experimental paradigm was adapted from a well-established task integrating emotion regulation and cognitive control processes [29]. It featured five trial types: passive viewing of neutral images, passive viewing of negative images, cognitive reappraisal of negative images, expressive suppression of negative images, and a digit-based Stroop task. Within each trial, emotion regulation and cognitive control components were interleaved, with participants first completing an emotion regulation task followed by a Stroop task requiring rapid cognitive evaluation. Each trial began with a fixation cross (“+”) displayed for 2000 ms, followed by a task cue presented for 2000 ms that instructed participants to view, reappraise, or suppress their emotional response passively. An emotional image (neutral or negative) was then shown for 3000 ms, followed by a jittered inter-stimulus interval of 800–1500 ms. Next, participants performed a digit-based Stroop task involving incongruent stimulus attributes (numerical magnitude vs. physical size). They were shown two digits (0–9) for 2000 ms and asked to identify the digit with the greater numerical value, ignoring physical size, responding with the “F” key if the left digit was larger and the “J” key if the right was larger. Task instructions varied by condition: in passive viewing (neutral/negative), participants observed the image and allowed any emotional response to occur; in cognitive reappraisal, they reinterpreted the negative image from a more optimistic perspective; in expressive suppression, they inhibited outward emotional reactions and maintained a neutral expression. Before the formal experiment, participants received standardized training sessions to familiarize themselves with the emotion regulation strategies (cognitive reappraisal and expressive suppression). During these training sessions, experimenters provided detailed explanations, demonstrations, and practice trials with immediate feedback, ensuring participants correctly understood and applied each strategy. This training aimed to minimize potential improvements in the post-test attributable solely to task familiarity rather than the intervention effect. After completing all task blocks, participants entered a picture rating phase, during which images were presented one at a time in random order. They rated each image for valence and arousal, enabling the assessment of emotional impact without interfering with the primary Stroop task. The trial structure of the image rating task is depicted in Figure 3.

#### 2.2.2. EEG Recording and Data Acquisition

EEG signals were recorded using a Neuroscan 64-channel system (Washington, D.C. USA), with AFz as the ground electrode and FCz as the reference. The data were band-pass filtered between 0.05 and 30 Hz, and a notch filter at 48–52 Hz was applied to eliminate line noise. Signals were sampled at 1000 Hz per channel, and electrode impedances were kept below 5 kΩ. Offline EEG analysis was conducted using Curry 8 software. Ocular artifacts were removed via Independent Component Analysis (ICA), and segments with amplitudes exceeding ±100 μV were excluded. Epochs were extracted based on task stages: for the image viewing and emotion regulation phase, epochs were time-locked to image onset and ranged from −200 to 1800 ms; for the Stroop task, epochs ranged from −100 to 1000 ms. Based on previous studies and grand-averaged ERP topographies, regions of interest (ROIs) and ERP components were defined. In the emotion regulation phase, the ROIs included CP1, CPz, CP2, P1, Pz, and P2, with the P3 component (300–390 ms) selected for analysis. In the Stroop task, ROIs included P3, Pz, and P4, focusing on the P3 component (300–390 ms), with mean amplitude as the metric.

### 2.3. Statistical Analysis

Statistical analysis was conducted using SPSS 26.0. A mixed-design ANOVA was performed for the emotion regulation phase, with 2 (group: experimental vs. control) × 2 (time: pre-test vs. post-test) × 4 (condition: neutral viewing, negative viewing, cognitive reappraisal, expressive suppression) factors. For the Stroop task, a repeated-measures ANOVA was conducted with the same group, condition, and time factors. Sphericity was tested using Mauchly’s test. In cases of violated sphericity, the Greenhouse–Geisser correction was applied. For significant main effects or interactions, post hoc comparisons were conducted using Tukey’s HSD for the mixed-design ANOVA, and Bonferroni corrections were applied to control for Type I error across multiple comparisons.

## 3. Results

### 3.1. The Effect of Physical Exercise on the Post-Emotional Adjustment Effect of High Psychological Stress in College Students

#### 3.1.1. Subjective Mood Ratings After Emotion Regulation

A repeated-measures ANOVA was performed on valence ratings across the four emotion regulation conditions. The assumption of sphericity was met (*p* > 0.05). A significant main effect of condition was found, F(3,135) = 11,081.45, *p* < 0.001, η^2^_p_ = 0.921, indicating substantial differences in valence ratings across conditions. Post hoc comparisons with Bonferroni correction revealed the following order of ratings: neutral viewing > negative viewing > expressive suppression > cognitive reappraisal (Figure 4).

No significant main effects were observed for group (F(1,45) = 1.81, *p* > 0.05, η^2^_p_ = 0.039) or time (F(1,45) = 0.005, *p* > 0.05, η^2^_p_ = 0.001). Likewise, there were no significant group × condition (F(3,135) = 1.35, *p* > 0.05, η^2^_p_ = 0.086) or group × time (F(1,45) = 0.44, *p* > 0.05, η^2^_p_ = 0.010) interactions. However, a significant condition × time interaction was observed, F(3,135) = 3.81, *p* < 0.05, η^2^_p_ = 0.210, along with a significant three-way group × condition × time interaction, F(3,135) = 6.50, *p* < 0.001, η^2^_p_ = 0.312.

Follow-up analyses revealed several condition- and group-specific effects. In the negative viewing condition, the experimental group had higher valence ratings pre-intervention (M = 3.404) than the control group (M = 3.366) but lower ratings post-intervention (M = 3.346 vs. 3.385). In the expressive suppression condition, post-test valence ratings were lower in the experimental group (M = 3.494) than in the control group (M = 3.520). Within the control group, valence ratings during cognitive reappraisal decreased slightly from pre- (M = 3.235) to post-test (M = 3.208). In the experimental group, valence ratings increased from pre- to post-test during neutral viewing (5.179 to 5.262) while decreasing during both negative viewing (3.404 to 3.346) and expressive suppression (3.524 to 3.494).

A repeated-measures ANOVA was conducted to examine the effects of emotion regulation condition, group, and time on arousal ratings. Results revealed a significant main effect of condition, F(3,135) = 74161.67, *p* < 0.001, η^2^_p_ = 0.892. Post hoc comparisons indicated the following ordered pattern: neutral viewing < cognitive reappraisal < expressive suppression < negative viewing. The main effect of group was also significant, F(1,45) = 37.11, *p* < 0.05, η^2^_p_ = 0.452, with overall higher arousal reported in the control group (M = 5.101) compared to the experimental group (M = 5.083).

Similarly, the main effect of time was significant, F(1,45) = 13.01, *p* < 0.05, η^2^_p_ = 0.224, showing a slight increase in arousal ratings from pre-test (M = 5.077) to post-test (M = 5.107). A significant condition × group interaction was observed, F(3,135) = 8.34, *p* < 0.05, η^2^_p_ = 0.368. Post hoc analyses revealed comparable within-group trends: control group: neutral viewing (M = 4.006) < cognitive reappraisal (M = 5.349) < expressive suppression (M = 5.493) < negative viewing (M = 5.558); experimental group: neutral viewing (M = 4.007) < cognitive reappraisal (M = 5.326) < expressive suppression (M = 5.463) < negative viewing (M = 5.536). The group × time interaction was also significant, F(1,45) = 30.63, *p* < 0.05, η^2^_p_ = 0.405; control group: arousal increased from pre-test (M = 5.063) to post-test (M = 5.140); experimental group: arousal decreased from pre-test (M = 5.091) to post-test (M = 5.075). A significant condition × time interaction was found, F(3,135) = 8.95, *p* < 0.05, η^2^_p_ = 0.384. Simple effects tests indicated the following: in the neutral viewing condition: pre-test (M = 3.968) < post-test (M = 4.045); in the negative viewing condition: pre-test (M = 5.522) < post-test (M = 5.571); in the cognitive reappraisal condition: pre-test (M = 5.332) < post-test (M = 5.343); in the expressive suppression condition: pre-test (M = 5.486) > post-test (M = 5.470). No significant three-way interaction between condition, group, and time was observed, F(3,135) = 1.75, *p* > 0.05, η^2^_p_ = 0.109.

These results suggest that arousal responses varied significantly across emotion regulation conditions and that they were modulated over time and between groups, particularly following intervention in the experimental group (see Table 1, Figure 5).

#### 3.1.2. Influence of Physical Exercise on Cognitive Control Interference Effects After Emotion Regulation in Highly Psychologically Stressed College Students

A repeated-measures ANOVA was conducted to investigate the effects of group (experimental vs. control), emotion regulation condition (neutral viewing, negative viewing, cognitive reappraisal, expressive suppression), and time (pre-test vs. post-test) on Stroop interference scores. The analysis included assessments of main effects and interaction effects.

A significant main effect of group was observed, F(1,45) = 5.31, *p* < 0.05, η^2^_p_ = 0.110, indicating that the experimental and control groups differed significantly in overall interference performance. However, neither the main effect of emotion regulation condition, F(3,135) = 1.31, *p* > 0.05, η^2^_p_ = 0.030 nor the main effect of time, F(1,45) = 2.03, *p* > 0.05, η^2^_p_ = 0.045, reached statistical significance. Further, the two-way interactions were non-significant: emotion regulation × group: F(3,135) = 0.86, *p* > 0.05, η^2^_p_ = 0.002; time × group: F(1,45) = 1.41, *p* > 0.05, η^2^_p_ = 0.032; emotion regulation × time: F(3,135) = 2.63, *p* > 0.05, η^2^_p_ = 0.058. Critically, a significant three-way interaction among emotion regulation condition, time, and group was found: F(3,135) = 9.80, *p* < 0.05, η^2^_p_ = 0.186. This result indicates that the temporal changes in Stroop interference varied across different emotion regulation strategies between the experimental and control groups (see Table 2 for detailed means and standard deviations).

To further examine the specific impact of the exercise intervention under different emotion regulation conditions, repeated-measures analyses were conducted separately for the experimental and control groups.

In the control group, no significant pre- to post-intervention changes in Stroop interference effects were observed under any of the emotion regulation conditions: neutral viewing: F(1,23) = 0.13, *p* > 0.05, η^2^_p_ = 0.003; negative viewing: F(1,23) = 2.10, *p* > 0.05, η^2^_p_ = 0.044; cognitive reappraisal: F(1,23) = 0.96, *p* > 0.05, η^2^_p_ = 0.021; expressive suppression: F(1,23) = 1.08, *p* > 0.05, η^2^_p_ = 0.023.

In contrast, the experimental group exhibited significant reductions in interference scores across all emotion regulation conditions, particularly under the following: neutral viewing: F(1,23) = 7.80, *p* < 0.05, η^2^_p_ = 0.148; negative viewing: F(1,23) = 22.12, *p* < 0.05, η^2^_p_ = 0.330. These findings suggest that the 12-week exercise intervention effectively reduced cognitive interference among high-stress college students, particularly in response to emotionally neutral and negative stimuli.

Furthermore, a significant overall effect of time across emotion regulation conditions was observed in the experimental group, F(3,69) = 3.70, *p* < 0.05, η^2^_p_ = 0.212. In contrast, no such effect was detected in the control group, F(3,69) = 0.36, *p* > 0.05, η^2^_p_ = 0.025.

Notably, between-group differences in reaction time under each emotional condition were not statistically significant after the intervention: neutral viewing: F(1,46) = 1.12, *p* > 0.05, η^2^_p_ = 0.075; negative viewing: F(1,46) = 2.61, *p* > 0.05, η^2^_p_ = 0.161. These results indicate that while the exercise intervention improved interference control in the experimental group, its effect on reaction times under various emotional conditions was less pronounced, suggesting a condition-specific and domain-specific modulation of cognitive performance (see Figure 6).

A repeated-measures ANOVA was conducted to examine the main and interaction effects of group (experimental vs. control), emotion regulation condition, and time (pre- vs. post-intervention) on the Stroop accuracy interference scores. As shown in Table 3, the main effect of group was significant, F(1,45) = 7.598, *p* < 0.05, η^2^_p_ = 0.150, indicating that participants in the experimental group exhibited significantly lower accuracy interference than those in the control group. The emotion regulation condition’s main effect was insignificant, F(3,135) = 0.997, *p* > 0.05, η^2^_p_ = 0.023, suggesting no overall differences in accuracy interference among the different emotion regulation strategies.

The main effect of time was significant, F(1,45) = 4.356, *p* < 0.05, η^2^_p_ = 0.092, with participants showing reduced interference following the 12-week intervention. However, none of the two-way or three-way interaction effects reached statistical significance: group × emotion regulation: F(3,135) = 0.780, *p* > 0.05, η^2^_p_ = 0.018; group × time: F(1,45) = 2.984, *p* > 0.05, η^2^_p_ = 0.065; emotion regulation × time: F(3,135) = 1.336, *p* > 0.05, η^2^_p_ = 0.030; group × emotion regulation × time: F(3,135) = 0.318, *p* > 0.05, η^2^_p_ = 0.007.

These results suggest that while the exercise intervention significantly improved accuracy-based interference control at a general level (particularly in the experimental group), the modulation of this effect did not vary by emotion regulation condition or over time between groups (see Figure 7).

### 3.2. Electrophysiological Effects of Physical Exercise on Emotion Regulation in Highly Psychologically Stressed College Students

EEG data were recorded using the Neuroscan wireless EEG recording system from the experimental and control groups at two time points (pre- and post-intervention). Event-related potentials (ERPs) were obtained through standard preprocessing and epoch-based signal averaging, applying a grand-average method to isolate component-level responses of interest.

Based on the waveform morphology and existing literature, the P3 component was identified during the Stroop task, reflecting emotional and cognitive control processes. For the P3 component, relevant to cognitive control, three parietal electrodes were selected: P3, Pz, and P4. Component-specific time windows were determined based on the grand-averaged ERP waveforms and peak latencies: Topographical voltage maps were generated within each component’s time window to visualize the scalp distribution of neural activity (see Figure 8). These topographic maps were then qualitatively assessed to confirm component localization and to examine potential group- and time-related differences.

To examine the effect of physical exercise on cognitive control following emotion regulation, P3 amplitudes during the Stroop task were analyzed using a repeated-measures ANOVA. The interference score was calculated as the difference between incongruent and congruent P3 amplitudes. The assumption of sphericity was met (*p* > 0.05). A significant main effect of Stroop condition was found, F(1,45) = 4.631, *p* < 0.05, η^2^_p_ = 0.106, with post hoc analysis indicating that P3 amplitudes were significantly larger in the congruent condition (M = 3.194 μV) than in the incongruent condition (M = 2.947 μV).

The main effect of group was also significant, F(1,45) = 7.616, *p* < 0.05, η^2^_p_ = 0.163. Post hoc comparisons revealed that the experimental group (M = 4.035 μV) exhibited significantly greater P3 amplitudes than the control group (M = 2.107 μV), suggesting enhanced cognitive control following the exercise intervention. No significant main effect of time was found, F(1,45) = 1.212, *p* > 0.05, η^2^_p_ = 0.030.

Importantly, a significant interaction between Stroop condition and group was observed, F(1,45) = 4.836, *p* < 0.05, η^2^_p_ = 0.110. Post hoc analysis showed that the difference in P3 amplitudes between congruent and incongruent conditions was significant in the experimental group (congruent > incongruent) but not in the control group, indicating that exercise enhanced the neural differentiation between congruent and incongruent stimuli. No significant interactions were found for condition × time: F(1,45) = 0.245, *p* > 0.05, η^2^_p_ = 0.006; group × time: F(1,45) = 0.580, *p* > 0.05, η^2^_p_ = 0.015; or condition × group × time: F(1,45) = 1.314, *p* > 0.05, η^2^_p_ = 0.033. Analysis of the P3 interference scores (incongruent minus congruent) further supported these findings. The control group exhibited significantly larger interference scores than the experimental group. Within the experimental group, interference scores were significantly reduced post-intervention compared to pre-intervention (*p* < 0.05), indicating improved cognitive conflict resolution following exercise (see Figure 9).

## 4. Discussion

### 4.1. The Effectiveness and Efficacy of Physical Exercise on Cognitive Reappraisal and Expressive Inhibition Emotion Regulation

By comparing subjective ratings of valence and arousal across four emotion regulation conditions—viewing neutral, viewing negative, cognitive reappraisal of negative images, and expressive suppression of negative stimuli—before and after an exercise intervention, we explored the influence of physical exercise on emotional experience. In this context, lower valence scores indicate reduced pleasantness, while higher arousal scores reflect greater emotional intensity.

As shown in Table 1, both the experimental and control groups exhibited a consistent pattern in valence: viewing neutral > viewing negative > expressive suppression > cognitive reappraisal. For arousal, the pattern was as follows: viewing neutral < cognitive reappraisal < expressive suppression < viewing negative. These results align with prior findings supporting the effectiveness of cognitive reappraisal and expressive suppression in modulating emotional experience. However, the present study also reveals nuanced differences between these two strategies.

Specifically, regarding valence, the experimental group reported significantly lower scores (i.e., less pleasantness) under expressive suppression than the control group, particularly post-intervention. In terms of arousal, the control group showed a trend of increasing arousal scores from pre- to post-test, whereas the experimental group showed a decreasing trend. These findings suggest that physical exercise may attenuate negative emotional experiences during expressive suppression, enhancing positive emotional states. Exercise may help individuals reframe emotionally charged events more positively, reducing the emotional burden of suppressing negative emotions. This, in turn, could make suppression more manageable and enhance subjective well-being, especially among individuals with high psychological stress [30,31]. Physical exercise supports emotion regulation via expressive suppression, helping individuals maintain positive affect and reduce psychological distress. This aligns with previous research suggesting that physical activity promotes psychological resilience and hedonic balance through mechanisms of emotional regulation [32,33].

While prior studies have primarily focused on the differential effects of reappraisal and suppression on emotional outcomes, our results further indicate the following: The experimental group showed significantly lower valence and arousal scores during expressive suppression post-intervention compared to their pre-intervention scores. During cognitive reappraisal, the experimental group displayed significantly increased arousal after the intervention, indicating heightened emotional engagement or regulatory effort. These findings provide compelling evidence that 12 weeks of physical exercise can effectively modulate subjective emotional responses across different emotion regulation strategies, with particular benefits observed in expressive suppression.

### 4.2. Effects of Physical Exercise on Cognitive Control Following Cognitive Reappraisal and Expressive Inhibition of Emotion Regulation

#### 4.2.1. Behavioral Outcomes

Under the congruent condition of the digit task, a significant interaction was observed between physical exercise and emotion regulation among college students with high psychological stress. Before the exercise intervention, the control group exhibited longer reaction times under the cognitive reappraisal condition than the expressive suppression condition. In contrast, no significant difference was found in the experimental group at pre-test. After the 12-week exercise intervention, neither group showed significant differences in reaction time between reappraisal and suppression under congruent conditions.

However, the interaction effect between physical exercise and emotion regulation was significant under the incongruent condition. Following the intervention, the experimental group demonstrated reduced reaction time during cognitive reappraisal, suggesting improved regulatory efficiency. Further comparison revealed that, before the intervention, the experimental and control groups had exhibited longer reaction times under reappraisal than suppression. Post-intervention, this difference remained in the control group but disappeared in the experimental group, indicating that exercise attenuated the disparity between the two strategies.

These findings suggest that the Stroop paradigm’s incongruent condition introduces automatic interference with the digit-size task, consistent with previous research [34]. Participants must inhibit this interference to accurately judge numerical magnitude, thus requiring greater cognitive resources. In contrast, the congruent condition imposes a lower cognitive load, and, therefore, emotion regulation strategies have limited behavioral impact under such conditions, regardless of exercise intervention. The results indicate that for students with high psychological stress, in low-load (congruent) conditions, physical exercise has no significant impact on cognitive control during emotion regulation. In high-load (incongruent) conditions, exercise facilitates cognitive control, particularly under the cognitive reappraisal strategy. This supports the hypothesis that physical exercise enhances emotion regulation under cognitive strain, allowing individuals to more efficiently engage with and regulate negative emotional stimuli. In particular, exercise reduces the cognitive cost of reappraisal, enabling individuals to cope with emotional conflict more flexibly and adaptively in high-demand situations.

The experimental group demonstrated significantly lower reaction time interference scores than the control group, and a significant reduction in reaction time interference scores was observed post-intervention across all four emotion regulation conditions (viewing neutral, viewing negative, cognitive reappraisal of negative, and expressive suppression of negative). In contrast, the control group showed no significant changes before and after the intervention. Lower reaction time interference scores indicate that response times during incongruent trials are closer to those in congruent trials, suggesting more efficient conflict control under cognitive–emotional interference.

Regarding accuracy interference scores, the experimental group showed significantly higher values than the control group, and the accuracy interference scores increased significantly after the exercise intervention. Higher accuracy interference scores imply that participants performed better under incongruent conditions than congruent ones, further reflecting enhanced cognitive conflict resolution ability. These findings suggest that physical exercise improves behavioral performance in emotion regulation tasks under cognitive conflict and enhances cognitive control in individuals under high psychological stress. The results are consistent with prior research showing that high levels of negative emotion impair self-regulatory capacity [35]. In real-life situations, individuals under negative emotional states often experience failures in self-control, potentially resulting in extreme maladaptive behaviors such as aggression, social withdrawal, or even suicidal tendencies [36,37].

Following the exercise intervention, participants in the experimental group exhibited greater cognitive control capacity between groups (vs. control) and within groups (pre- vs. post-intervention), particularly under the cognitive reappraisal and expressive suppression conditions during the Stroop task. These improvements suggest that exercise reduces the cognitive cost of emotion regulation, especially under high-stress emotional demands. This can be explained from a neurocognitive perspective. Humans are evolutionarily wired to avoid rather than accept negative emotional stimuli [38]. Therefore, emotion regulation strategies—regardless of type—often counteract instinctual responses, making the regulation process inherently effortful and resource-consuming. Under cognitive resource depletion or hyper-arousal conditions, successful emotion regulation becomes even more difficult [39].

According to James Papez’s theory of the Papez circuit, key structures involved in emotion regulation include the hypothalamus, cingulate gyrus, hippocampus, and anterior thalamus [40]. Emotion-related neurotransmitters such as serotonin, dopamine, and norepinephrine are deeply involved in the modulation of emotional states. Exercise-induced improvements in emotion regulation and subsequent enhancements in cognitive control may be linked to increased levels of these neurotransmitters, as supported by extensive empirical evidence [41]. However, it is important to note that participating in structured group activities, such as exercise programs, may also offer psychological and social benefits, such as social interaction, routine, and stress relief, which could contribute to improvements in emotional regulation and cognitive control [42].

Furthermore, regular physical exercise strengthens implementation intentions—specific, goal-directed action plans—which have been shown to reduce cognitive load during emotional processing [43]. The potential psychological and social aspects of group exercise, such as the routine and the social support, could also play a role in reducing cognitive load and promoting emotional regulation alongside the physiological effects of exercise [44]. When implementation intentions guide regulation, the activation of emotion-related networks (including cognitive control, memory retrieval, and motor preparation circuits) is significantly reduced, compared to spontaneous emotion regulation [45]. These benefits may stem not solely from the physical aspects of exercise but also from the structured and supportive environment created through social interaction and routine, which can aid in emotional regulation. Stronger connectivity within these networks is associated with greater emotional regulation difficulty, suggesting that implementation intentions can make regulating negative emotions less effortful and more automatic.

#### 4.2.2. P3 Amplitude

This study represents the first neurophysiological investigation into how physical exercise influences cognitive control following emotion regulation in individuals with high psychological stress. Specifically, we examined the P3 amplitude during a Stroop task conducted after applying two emotion regulation strategies—cognitive reappraisal and expressive suppression—to assess how these strategies modulate the cognitive resources required for subsequent task performance. The P3 component is commonly associated with attentional processes, but broader neurophysiological mechanisms, including neurotransmitter activity (e.g., dopamine and norepinephrine), overall arousal state, and neuronal resource allocation, also influence its amplitude [46]. A repeated-measures ANOVA on P3 amplitudes revealed a significant main effect of condition: P3 amplitudes were significantly larger in congruent trials than in incongruent trials. This finding aligns with previous studies, indicating that congruent trials are less affected by inhibitory control demands. In contrast, incongruent trials involve greater cognitive inhibition, leading to reduced P3 amplitudes due to increased resource allocation.

Importantly, following emotion regulation, individuals tend to experience cognitive resource depletion, resulting in a preference for simpler decision-making strategies in Stroop tasks. Such depletion is also linked to shifts in neurotransmitter levels and altered arousal states, influencing neuronal efficiency and resource allocation [46]. These neurochemical and physiological changes are particularly pronounced under chronic stress conditions, where disruptions in neurotransmitters like dopamine and serotonin significantly impair cognitive performance and emotional regulation [47]. Consequently, the brain’s ability to efficiently allocate resources toward critical cognitive operations, such as memory consolidation, attentional focus, and executive decision-making, is compromised. This depletion phenomenon clarifies why cognitive control demands—reflected in P3 amplitudes—are less impacted during congruent compared to incongruent conditions. Essentially, simpler tasks require fewer cognitive resources, thus remaining relatively unaffected by depleted states. Additionally, a significant main effect of group was observed, with the experimental group that had engaged in the physical exercise intervention exhibiting greater cognitive control post-emotion regulation compared to the control group. This finding strongly supports the notion that physical exercise contributes positively to cognitive resource availability, potentially by stabilizing neurotransmitter levels and maintaining optimal arousal states. Therefore, an exercise intervention not only facilitates better emotion regulation but also enhances subsequent executive functioning and overall cognitive resilience [48].

Furthermore, the interaction between condition and group was significant. In the control group, no significant difference in P3 amplitudes was found between congruent and incongruent trials. In contrast, the experimental group exhibited significantly larger P3 amplitudes during congruent trials than during incongruent trials. This reinforces the conclusion that exercise enhances post-regulatory cognitive control, especially under conditions that require less inhibition. To further assess cognitive load, P3 interference scores were calculated (i.e., the difference between congruent and incongruent P3 amplitudes). Lower interference scores indicate that the P3 amplitude in incongruent trials approaches that of congruent trials, reflecting reduced cognitive disruption. In this study, the control group exhibited significantly higher P3 interference scores than the experimental group. Moreover, the experimental group’s interference scores decreased significantly post-intervention, while the control group showed no such change.

These results can be interpreted within the framework of the process-specific timing hypothesis, which posits that the later emotion regulation occurs during emotional processing, the more cognitively demanding it becomes [49]. Short-lived or weak emotional stimuli may be filtered out with minimal cognitive cost. In contrast, chronic negative emotions, often present in individuals under prolonged stress, require more substantial cognitive resources to regulate due to the breakdown or adaptation of earlier filtering mechanisms.

Participants in the control group, likely exposed to prolonged psychological stress, may have accumulated unresolved negative emotions, leading to elevated P3 interference scores. By contrast, although the experimental group started with similar stress exposure, regular physical exercise effectively reduced negative emotional experiences and facilitated a more adaptive emotion-to-cognition transition. However, it is important to consider that participating in a structured group activity may also bring psychological and social benefits, such as social interaction, routine, and stress relief, which could contribute to improved emotional regulation and more adaptive cognitive processing [44]. Exercise may have induced more stable and positive affective states, allowing participants to conserve cognitive resources during emotion regulation and subsequent task performance. These benefits may stem from the physiological effects of exercise and the psychological and social aspects of group participation, which can enhance emotional resilience and cognitive performance.

Taken together, both the P3 amplitude patterns and P3 interference scores suggest that physical exercise serves a neurocognitive protective role, reducing the impact of emotional regulation on cognitive resource depletion and enhancing executive control in high-stress populations. Nonetheless, the nonspecific effects of participating in a group activity, such as routine and social support, should also be considered as plausible contributors to these improvements.

### 4.3. Limitations

While the current study provides valuable insights into the positive effects of physical exercise on emotional regulation and cognitive control, several limitations must be acknowledged. First, this study’s self-selection bias is inherent, as only participants who volunteered for the exercise intervention were included. These individuals may be more motivated or already more physically active, which could influence the generalizability of the findings. This self-selection bias may impact the internal validity of the results, and the conclusions drawn from this sample may not apply to less active or less motivated populations.

In addition, several methodological constraints limit the generalizability and interpretability of the findings. The sample in this study consisted solely of Chinese college students, which restricts the ability to generalize the results to other populations. Furthermore, adherence to the exercise regimen was not directly monitored, leaving the possibility that some participants did not consistently meet the intended exercise intensity or duration. Additionally, the lack of control for time-of-day effects during testing is another limitation, as circadian influences may have impacted cognitive performance and emotional regulation. The absence of a follow-up assessment is another important limitation, as the long-term effects of exercise on emotional regulation and cognitive control remain unclear. Finally, potential confounders, such as sleep quality, nutrition, and stress levels, were not accounted for in this study, yet these factors could have influenced the observed outcomes.

## 5. Conclusions

Both expressive suppression and cognitive reappraisal were effective strategies for emotion regulation, with cognitive reappraisal demonstrating superior regulatory effects. Psychological stress often triggers elevated levels of negative emotions, weakening individuals’ behavioral control and impairing their ability to exert cognitive control in subsequent conflict tasks. Moreover, regulating emotions may further deplete the cognitive resources available for later tasks.

However, physical exercise appears to offer a protective benefit in this context, modestly reducing the adverse effects of emotion regulation on post-regulatory cognitive control among individuals experiencing high psychological stress. Nevertheless, due to the multifaceted nature of the exercise intervention—which included elements such as social engagement, structured routine, and distraction—these improvements cannot be solely attributed to physical activity alone. It is likely that nonspecific components of the intervention also contributed to the observed cognitive resilience and executive functioning improvements. Future research should aim to isolate the specific mechanisms through which physical exercise and other intervention factors jointly enhance cognitive and emotional regulation.

## Figures and Tables

**Figure 1 healthcare-13-02100-f001:**
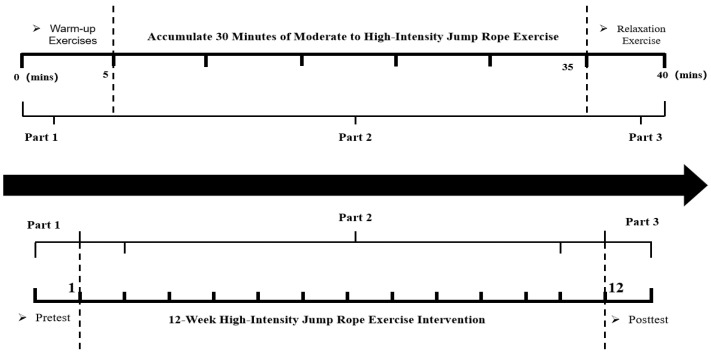
The 12-week high-intensity rope-skipping exercise intervention. Ps: The top panel represents the time of each intervention (5 min of warm-up (part 1), 30 min of high-intensity jump rope exercise (part 2), and 5 min of cool-down exercises (part 3)); the bottom panel illustrates the 12-week intervention schedule (pretest (part 1), 12-week intervention (part 2), and post-test (part 3)).

**Figure 2 healthcare-13-02100-f002:**
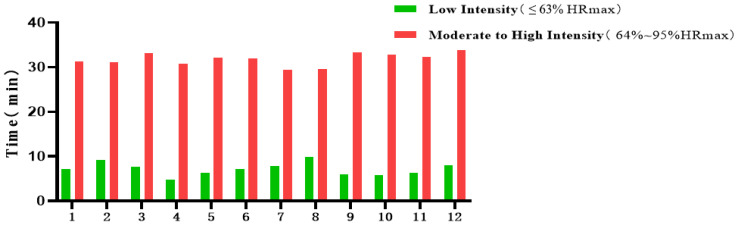
Intervention monitoring of physical exercise for 12 weeks.

**Figure 3 healthcare-13-02100-f003:**
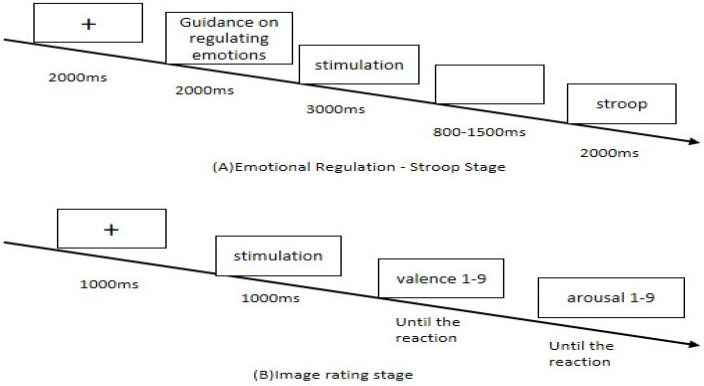
Emotion regulation tasks.

**Figure 4 healthcare-13-02100-f004:**
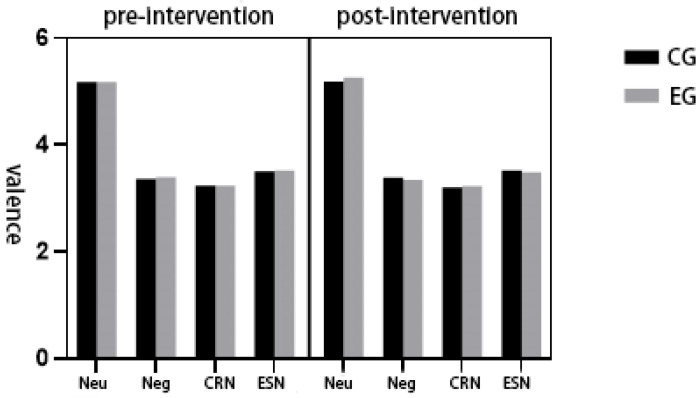
Subjective experience ratings of both groups of subjects before and after the physical exercise intervention in the four emotion regulation conditions.

**Figure 5 healthcare-13-02100-f005:**
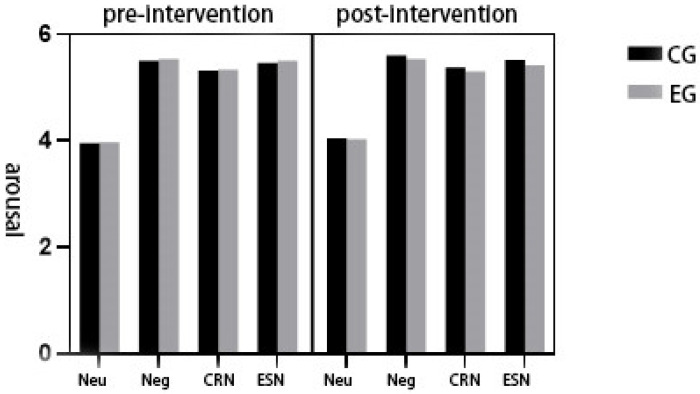
Subjective arousal ratings before and after the physical exercise intervention in both groups of subjects in the four emotion regulation conditions.

**Figure 6 healthcare-13-02100-f006:**
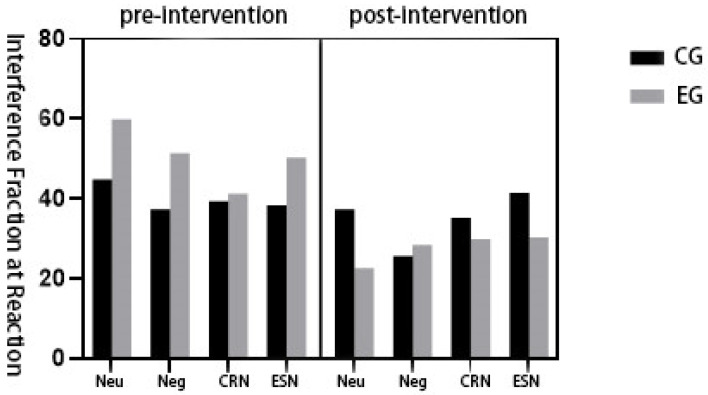
Changes in interference scores during emotion regulation responses by physical exercise interventions for highly psychologically stressed college students.

**Figure 7 healthcare-13-02100-f007:**
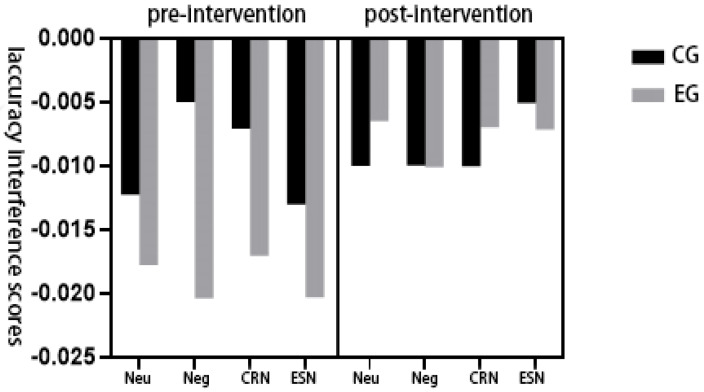
Changes in the interference scores of physical exercise interventions on the correct rate of emotion regulation of highly psychologically stressed university students.

**Figure 8 healthcare-13-02100-f008:**
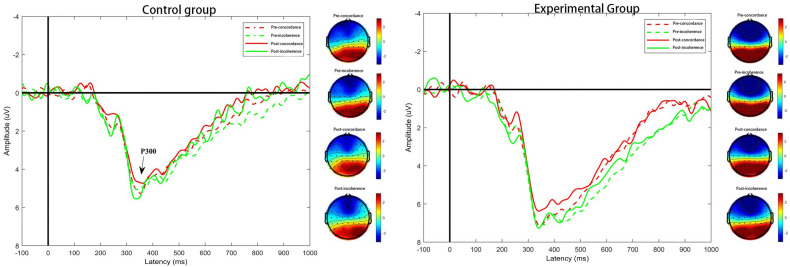
Mean superimposed waveforms and topographic maps of the control and experimental groups (P3).

**Figure 9 healthcare-13-02100-f009:**
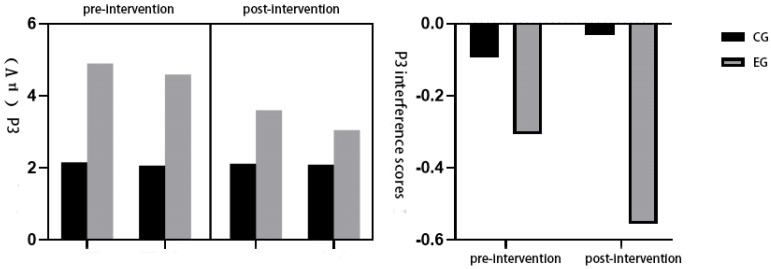
Changes in P3 after physical exercise intervention on emotion regulation in highly psychologically stressed college students.

**Table 1 healthcare-13-02100-t001:** Subjective experience ratings of both groups of subjects before and after the physical exercise intervention in the four emotion regulation conditions.

Groups		Emotional Conditioning			
		Watch Neutral	Watch Negative	Cognitive Reassessment Negative	Expression Inhibition Negative
pre-intervention	control group				
	valence	5.180	3.366	3.235	3.506
	arousal	3.958	5.507	5.320	5.466
	experimental group				
	valence	5.179	3.404	3.241	3.524
	arousal	3.978	5.537	5.343	5.506
post-intervention	control group				
	valence	5.191	3.385	3.208	3.520
	arousal	4.054	5.608	5.377	5.521
	experimental group				
	valence	5.262	3.346	3.230	3.494
	arousal	4.036	5.535	5.308	5.420

**Table 2 healthcare-13-02100-t002:** The influence of physical exercise on the interference effect of cognitive control after emotion regulation in highly psychologically stressed college students (reaction time).

	Type III Sum of Squares	df	Mean Square	*F*	*p*	*η* ^2^ _p_
Category	2087.971	3	695.99	1.313	0.273	0.03
groups	13,990.44	1	13,990.44	5.307	0.026	0.11
Type × group	1371.388	3	457.129	0.862	0.463	0.02
times	20,589.98	1	20,589.98	2.03	0.161	0.045
Time × group	14,313.64	1	14,313.64	1.411	0.241	0.032
Category × Time	6128.642	3	2042.881	2.628	0.053	0.058
Category × Time × Group	3860.806	3	3860.806	9.803	0.003	0.186

**Table 3 healthcare-13-02100-t003:** The influence of physical exercise on the interference effect of cognitive control after emotion regulation in highly psychologically stressed college students (correct rates).

	Type III Sum of Squares	df	Mean Square	*F*	*p*	*η* ^2^ _p_
Category	0.002	3	0.002	0.997	0.396	0.023
groups	0.011	1	0.011	7.598	0.009	0.15
Type × group	0.002	3	0.002	0.78	0.507	0.018
times	0.008	1	0.008	4.365	0.043	0.092
Time × group	0.006	1	0.006	2.984	0.091	0.065
Category × Time	0.003	3	0.003	1.336	0.266	0.03
Category × Time × Group	0.001	3	0.001	0.318	0.812	0.007

## Data Availability

Please contact the corresponding authors to obtain the data.

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
