# Peer review of "The Effects of Moderate- to High-Intensity Physical Exercise on Emotion Regulation and Subsequent Cognitive Control in Highly Psychologically Stressed College Students"

_healthcare, 2025, doi:10.3390/healthcare13172100_

Round 1
Reviewer 1 Report
Comments and Suggestions for Authors
Title – The Effects of Moderate to High Intensity Physical Exercise on Emotion Regulation and Subsequent Cognitive Control in Highly Psychologically Stressed College Students
General Comment:
This manuscript explores the effects of a 12-week moderate-to-high intensity rope-skipping intervention on emotion regulation and cognitive control among highly stressed college students. The study is timely and relevant, using a mixed-method approach combining behavioral data and ERP (P3) components. The experimental design is generally appropriate, and the findings may contribute to the literature on stress resilience and exercise-based interventions. However, several key issues regarding methodological transparency, reporting structure, and figure interpretation must be addressed before the manuscript is suitable for publication.
Line 111: Although the manuscript states that participants were “randomly assigned” to either the experimental or control group, the randomization method (e.g., computer-generated sequence, block randomization, gender stratification) is not described. Please provide details regarding the random allocation procedure to ensure reproducibility.
Line 141: The experimental procedures are currently scattered across multiple subsections. For clarity and replication purposes, please consolidate and present the design sequentially under a dedicated “Experimental Design” section, detailing the timeline and steps of (1) participant screening and grouping, (2) pretesting, (3) intervention implementation, and (4) post-testing, including EEG and behavioral tasks.
Line 156: The manuscript does not specify whether all pre- and post-test assessments were conducted at the same time of day, which may affect results due to circadian variation. Moreover, it is unclear whether the Chinese Affective Picture System (CAPS) and EEG recordings were conducted simultaneously or sequentially. Please clarify the timing and structure of these measurements.
Line 171: Please revise Figure 1 to use standardized and precise English terminology.
Line 196: It is unclear whether the 30-minute rope-skipping sessions involved continuous activity or were broken into intervals. Were there any work-to-rest ratios (e.g., 3 × 10 min with breaks)? Clarifying this aspect is essential for understanding the intensity and load of the intervention.
Line 211: Figure 2 includes labels such as “Part 1”, “Part 2”, and “Part 3”, but these segments are not explained in the figure legend or in the main text. Please clarify what each part represents (e.g., phases of the 12-week intervention or daily structure), and consider revising the figure for improved interpretability.
Line 235: More detailed information about the physical activity level of the control group should be provided. Information such as weekly course duration, intensity, content type should be specified. Otherwise, the difference in exercise intensity cannot be interpreted with certainty.
Line 239: Perceived exertion or mean heart rate data during exercise should be added or at least discussed as a limitation to comply with the ACSM protocol.
Line 263: The statistical analysis procedures are currently embedded under the EEG recording section. It would improve clarity to present a standalone “Statistical Analysis” subsection outlining the software used, ANOVA model designs, assumption checks, and post-hoc correction methods. Furthermore, although partial eta-squared values are reported, additional effect size metrics (e.g., Cohen’s d) for key comparisons would enhance the interpretation of findings.
Line 637: When interpreting the P3 component, only its relation to attentional processes is emphasized. Broader explanations are needed: e.g., neurotransmitter levels, arousal state, or neuronal resource allocation could be mentioned in the context of the significance of P3.
Line 704: The manuscript lacks a clearly defined “Limitations” section. Important methodological constraints—such as the limited generalizability of the sample (e.g., only Chinese college students), absence of detailed monitoring of adherence (e.g., unclear if all participants maintained target heart rates), lack of time-of-day control during testing (e.g., possible circadian effects), absence of follow-up assessment (e.g., no data on long-term impact), and omission of potential confounders (e.g., sleep, nutrition, stress)—are not sufficiently addressed. I recommend adding a concise limitations paragraph at the end of the Discussion to improve methodological transparency and interpretive clarity.
Reviewer 2 Report
Comments and Suggestions for Authors
Dear authors,
Thank you for the opportunity to review your manuscript. This is a timely and relevant study addressing the effects of physical exercise on emotion regulation and cognitive control among highly stressed college students. The study is well-conceived, presents a rich dataset, and offers promising findings.
Overall, the manuscript is clearly written and well-organized. However, several important aspects need further clarification or expansion. In particular, I encourage you to better define your principal variables, add detail to the methodological description, and broaden your discussion to acknowledge alternative interpretations of the results beyond the physiological effects of exercise alone.
Below, I provide specific comments by section to help guide your revisions. I believe that with these improvements, your manuscript will be significantly strengthened and well positioned for publication.
Introduction
- The concept of Subsequent Cognitive Control, which is a central variable in the study, is not clearly defined. Similarly, the criteria used to determine "high psychological stress" are not explained. The authors should clarify what constitutes high stress, its prevalence among university students, and how it impacts them psychologically and physiologically.
- Lines 85–98 provide insufficient justification for how physical exercise may influence emotion regulation and cognitive control. The authors should also consider including broader psychosocial mechanisms, such as psychological well-being, social engagement, and stress distraction, which may be just as or more influential.
- The rationale for selecting moderate-to-vigorous intensity physical activity is not provided. The introduction should briefly justify the chosen intervention parameters to contextualize the study appropriately.
- As a suggestion, the authors may consider stating their hypotheses early in the introduction to clarify their initial expectations and guide the reader.
Methods
2.1 Participants and Screening Procedure
- The stress assessment instrument (CCSPSS) is insufficiently described. Key details such as the number of items, sample questions, reliability (e.g., Cronbach’s alpha), and its validation history are missing. Include a brief description of the CCSPSS and its psychometric properties.
- Sampling procedures lack detail. It is unclear how the 715 students were approached (e.g., email, in-class recruitment) or whether any stratification by gender, age, academic field, or level occurred. Describe the sampling method and discuss whether the sample is representative of the high-stress college student population.
2.2.1 Materials
- The validation process for the image stimuli is underdescribed. It is not clear who the 10 evaluators were, whether they followed the same inclusion criteria, or how they were selected (randomly or otherwise).
2.2.2 Experimental Task and Procedure
- It is not specified whether participants received training in applying the emotion regulation strategies (reappraisal and suppression). Without this, there is a risk that improvements in the post-test may partly reflect familiarity with the task rather than the effect of the intervention.
2.2.3 Physical Activity Intervention
- A key sentence lacks citation: Lines 203-206:“Importantly, prior research indicates that low, moderate, or high-intensity aerobic exercise can all positively affect emotional well-being…” The authors should provide references to support these claims.
- The actual physical activity levels of the control group are not clearly defined. It is also unclear whether adherence to the “no additional exercise” instruction was monitored.
- Suggestion: In Figure 2, consider adding a subtitle indicating that the top panel represents the structure of each session and the bottom panel illustrates the 12-week intervention schedule.
Discussion
- The discussion attributes all positive outcomes to physical exercise (e.g., cardiovascular activation, neurotransmitter changes, increased P3 amplitudes), without considering other contributing factors. Participating in a structured group activity may bring psychological and social benefits (e.g., social interaction, routine, stress relief) that could also account for improvements. The authors should discuss these nonspecific effects as plausible alternative mechanisms and include this issue in the limitations.
- As stated in the Methods section, only participants who volunteered for the exercise intervention were included. This introduces potential self-selection bias, as these individuals might be more motivated or already more active. Acknowledge this bias explicitly in the discussion as it may impact the internal validity of the findings.
Conclusions
- The conclusions are generally well articulated. However, the authors should better qualify that the effects of physical exercise, while positive, appear to be modest. Given the multifaceted nature of the intervention (which includes social engagement, routine, and distraction), the improvements cannot be attributed solely to physical exercise. I recommend to soft the causal tone of the conclusions and acknowledge the possible influence of nonspecific intervention components.
Round 2
Reviewer 1 Report
Comments and Suggestions for Authors
The requested revisions for the manuscript have been appropriately addressed.